# Uterine Cavity Lavage Mutation Analysis in Lithuanian Ovarian Cancer Patients

**DOI:** 10.3390/cancers15030868

**Published:** 2023-01-30

**Authors:** Diana Žilovič, Ieva Vaicekauskaitė, Rūta Čiurlienė, Rasa Sabaliauskaitė, Sonata Jarmalaitė

**Affiliations:** 1Institute of Biosciences, Vilnius University, Sauletekio Avenue 7, LT-10222 Vilnius, Lithuania; 2Laboratory of Clinical Oncology, National Cancer Institute, Santariškių 1, LT-08406 Vilnius, Lithuania; 3Oncogynaecology Department, National Cancer Institute, Santariškių 1, LT-08406 Vilnius, Lithuania; 4Laboratory of Genetic Diagnostic, National Cancer Institute, Santariškių 1, LT-08406 Vilnius, Lithuania; 5National Cancer Institute, Santariškių 1, LT-08406 Vilnius, Lithuania

**Keywords:** uterine lavage, ovarian cancer, liquid biopsy, *TP53*

## Abstract

**Simple Summary:**

Ovarian cancer is the most lethal gynecological malignancy. The overall survival of patients this disease had not substantially changed for several decades, mainly due to the lack of early diagnosis. Type II OC is the most common and most aggressive form of OC, which mainly includes high-grade serous ovarian carcinoma (HGSOC). Our study aimed to pilot whether the detection of *TP53* mutation in uterine cavity lavage can be used as a diagnostic method for type II OC. Uterine lavage technique was successfully applied to all patients, also ovarian tissue biopsy was taken. All 136 samples (90 uterine cavity lavages and 46 tissues) were sequenced using six gene panel that included genes commonly associated with ovarian and endometrial cancers (*TP53*, *BRCA1*, *BRCA2*, *PIK3CA*, *KRAS*, and *PTEN*). Our pilot study proved that ctDNA from ovarian neoplasms can be collected from uterine lavage for diagnostic needs. We revealed precise detection of *TP53, BRCA1, BRCA2* in uterine lavage from HGSOC by means of NGS. However, for improved sensitivity of such test, additional disease-specific biomarkers have to be discovered.

**Abstract:**

Background: Type II ovarian cancer (OC) is generally diagnosed at an advanced stage, translating into a poor survival rate. Current screening methods for OC have failed to demonstrate a reduction in mortality. The uterine lavage technique has been used to detect tumor-specific *TP53* mutations from cells presumably shed from high-grade serous ovarian cancer (HGSOC). We aimed to pilot whether the detection of *TP53* mutation in uterine cavity lavage can be used as a diagnostic method for HGSOC using an expanded gene panel. Methods: In this study 90, uterine lavage and 46 paired biopsy samples were analyzed using next-generation sequencing (NGS) targeting *TP53* as well as five additional OC-related genes: *BRCA1*, *BRCA2*, *PI3KCA*, *PTEN*, and *KRAS*. Results: Uterine lavage was successfully applied to all patients, and 56 mutations were detected overall. *TP53* mutations were detected in 27% (10/37) of cases of type HGSOC; *BRCA1* and *BRCA2* mutations were also frequent in this group (46%; 17/37). Overall concordance between tissue and liquid biopsy samples was 65.2%. Conclusion: Uterine lavage *TP53* mutations in combination with other biomarkers could be a useful tool for the detection of lowly invasive HGSOC.

## 1. Introduction

Ovarian cancer (OC) is the most lethal gynecological malignancy. Due to the lack of early OC symptoms and effective screening approaches, approximately 60–70% of OC cases are diagnosed in advanced stages, with a 31% 5-year survival rate. In contrast, survival for women with localized disease is 92%, indicating that early OC detection could vastly decrease mortality [1,2]. OC is a highly heterogeneous disease, including different histological subtypes. A dualistic model of epithelial OC carcinogenesis based on different molecular and pathogenetic features was proposed by Kurman et al., distinguishing type I and type II OC. This model provides important insights into the origin of OC [3,4]. Type I OC is believed to develop in a stepwise manner from benign precursor lesions, notably borderline or atypical proliferative tumors. This OC type is usually diagnosed at an early stage as non-aggressive, low-grade tumors. Type I tumors include low-grade serous, endometrioid, clear-cell, mucinous carcinomas, and malignant Brenner tumors. Type II OC is the most common and most aggressive form of OC, which mainly includes high-grade serous ovarian carcinoma (HGSOC). HGSOC is diagnosed at advanced stages in ~70% of the cases. HGSOC is believed to arise in the fallopian tube epithelium, and mutations in the tumor suppressor gene *TP53* are assumed to be a very early event in the carcinogenesis of HGSOC. An important feature of HGSOC, which may facilitate early detection, is the high prevalence of tumor protein p53 gene (*TP53*) mutations (>96%), even in premalignant lesions [4,5,6,7,8,9].

A growing number of studies have revealed the involvement of multiple genes and pathways in the pathogenesis of OC; the frequency of the spectrum of mutations varies among different subtypes of epithelial OC [8,9,10,11,12,13,14]. Type I OC is characterized by mutations in genes such as *PIK3CA, KRAS, BRAF, MET, PTEN, ERBB2, ARID1A*, *CTNNB1, TERT, RPL22, RNF43,* and others, depending on their histological subtype. They rarely harbour *TP53* mutation and are relatively genomically stable. Type II OC are genomically unstable, have widespread copy number alterations, and ubiquitous *TP53* mutations (>96%). Other common threads include *CCNE1* amplification (20%), germline and somatic mutations of *BRCA1/2* (20–40%), and other aberrations in pathways of DNA damage response. In addition, mutations of *RB1* (9%), *NF1* (4–11%), *LRP1b* (8%), *PTEN* (6%), *CSMD3* (6%), *FAT3* (6%), *KRAS* (5%), *CREBBP* (5%), *WWOX* (4%), *ANKRD* (4%), *MAP2K4 (3%)*, and *PIK3CA* (2%) can be detected in type II OC [9,10,11,12,13,14].

Despite years of research, the diagnosis of early-stage cancer remains extremely challenging. In recent years, several studies [15,16,17,18,19,20] investigated cell-free circulating tumor DNA (ctDNA) in uterine lavage as a potential biomarker of OC. Uterine lavage techniques have been used to collect ctDNA shed from fallopian tubes, with the majority of reports looking for the *TP53* mutations [15,16,17,18,19,20].

In this pilot study, we aimed to assess whether the detection of *TP53* and other OC-specific mutations in uterine cavity lavage can be used as a diagnostic tool for HGSOC using targeted next-generation sequencing.

## 2. Materials and Methods

### 2.1. Patient Cohort

The patient cohort consisted of 90 patients who underwent surgery with pre-operative concern for an ovarian malignancy, uterine cancer, benign gynecological tumor or prophylactic salpingoovarectomy for the identified *BRCA1/2* germline mutation at the National Cancer Institute of Lithuania between 2018 and 2021. Overall, the patient cohort consisted of 37 patients with type II OC (HGSOC) and 53 patients with other gynecologic diseases: 9 patients with type I OC (1 clear-cell, 3 borderline, 2 mucinous, 2 with simultaneous endometrioid ovarian and endometrial malignant tumors, and 1 granulosa cell tumor), 12 patients with endometrial carcinoma, 19 cases with benign gynecologic tumors, and 13 *BRCA1/2* mutation carriers, who underwent risk-reducing surgery (RRS) for hereditary breast and ovarian cancer (including one patient with HGSOC precursor STIC—serous tubal intraepithelial carcinoma) (Figure 1). All participants were informed about the study and signed a written consent form. The study was approved by the Regional Bioethics Committee (No. 158200-18/5-988-539).

The clinical features of the patients and the pathological features of the tumour included in the study are provided in Table 1.

### 2.2. Uterine Cavity Lavage and Ovarian Tissue Sample Collection and DNA Extraction

Uterine lavage samples were successfully collected from 90 patients following a protocol under general anesthesia before surgery. An antiseptic lotion was used to clean the cervix. Using bullet forceps, the cervix was grasped, a two-way hysterosalpingography catheter was inserted into the cervical canal, and the balloon was inflated with approximately 2–3 mL of saline to seal the cervical canal and prevent retrograde leakage of saline. If the cervical canal was too narrow to pass the catheter, it was dilated to 2–3 mm with Hegar dilators. One 5-mL syringe containing 5 mL of saline was connected to the catheter tube. By pushing on the plunger of the syringe containing saline, the uterine cavity was slowly perfused. Then the syringe was gently pulled out and uterine lavage was collected. Finally, the balloon was deflated, and the catheter was removed.

Immediately following the collection procedure, the uterine lavage sample was centrifuged for 15 min at 2000× *g*. The resulting supernatant was discarded, and the cellular debris was washed with a 2 mL phosphate buffered saline (PBS) solution. The resulting uterine lavage cell pellet was resuspended in 2 mL PBS and stored at −80 °C until use.

1 mL of uterine lavage sample was used for DNA extraction using the Magmax^TM^ Cell-free Isolation Kit (Applied Biosystems, Thermo Fisher Scientific (TFS), Foster, CA, USA) following the manufacturer’s protocol. The final tissue and uterine cavity lavage DNA samples were stored at −20 °C until library preparation.

During surgery, a small sample of tumor tissue was allocated for analysis and immediately stored at −80 °C. 46 paired tissue and uterine lavage samples were collected for the analysis: 29 type II OC, 7 other ovarian tumors, 1 endometrial cancer, and 1 RRS group, 8 benign tumors. For genomic DNA extraction, the ovarian tissue samples were mechanically homogenised in liquid nitrogen using a mortar and pestle. 10–20 mg of tissue powder was digested with proteinase K (ThermoScientific, TFS, Vilnius, Lithuania) for 16 h, then genomic DNA was purified following standard phenol-chloroform extraction and ethanol precipitation protocols. The final DNA was dissolved in nuclease-free water (Invitrogen, TFS, Austin, TX, USA), and stored at −20 °C until further steps.

### 2.3. Targeted Next-Generation Sequencing

In all, 136 samples (90 uterine cavity lavages and 46 tissues) were sequenced using a custom-targeted six gene panel that included genes commonly associated with ovarian and endometrial cancers (*TP53*, *BRCA1*, *BRCA2*, *PIK3CA*, *KRAS*, and *PTEN*). DNA concentration was determined using the Qubit™ dsDNA HS Assay Kit on a Qubit™ 2.0 Fluorimeter (Invitrogen, TFS, Eugene, OR, USA). Up to 10 ng/sample of genomic DNA was used for library preparation using AmpliSeq™ Library Kit 2.0 and Ion AmpliSeqTM Custom DNA Panel (Life Technologies (LT), Carlsbad, CA, USA) according to the manufacturer’s directions. Ion Library TaqMan™ Quantification Kit (AB, TFS, Vilnius, Lithuania) was used for the sequencing library quantification. The next-generation sequencing was carried out using the Ion Torrent™ Ion S5™ system on Ion 530^TM^ chips. Data analysis was conducted automatically on the Ion Reporter 5.18 tool (LT, Carlsbad, CA, USA), where sequence reads were aligned to human reference genome 19 (Genome Reference Consortium GRCh37). Additionally, each alignment was visualized and verified using the Integrative Genomics Viewer 2.4.8 tool. All detected variants were classified according to American College of Medical Genetics and Genomics (ACMG) recommendations using the ClinVar (NCBI) database.

### 2.4. Statistical Analysis

Progression-free survival (PFS) was measured after one year of observation. The observation period started at the date of OC diagnosis and ended at the date of the first recurrence or death. Data analysis and graphical annotation were performed on Rx64 4.0.3 using RStudio 1.4.1717. Oncoprints are generated using the ComplexHeatmap package version 2.11.1. Uterine lavage mutation performance for type HGSOC cancer classification characteristics (sensitivity, specificity, accuracy, positive predictive value (PPV), negative predictive value (NPV), was also calculated using the ROCit (version 2.1.1) package. The uterine lavage mutation predictive value of HGSOC was assessed using risk ratio and odds ratio characteristics. Kaplan-Meier curves, univariate and multivariate Cox regression analysis were used to determine the association of uterine lavage mutation with progression-free survival (PFS). Results are considered statistically significant if the *p*-value < 0.050.

## 3. Results

### 3.1. Mutation Analysis in Uterine Cavity Lavage Samples

Mutation analysis of six OC-related genes (*TP53, BRCA1, BRCA2, PIK3CA, PTEN,* and *KRAS*) was first performed in 90 uterine lavage samples. 41 (45.6%) samples had at least one mutation, four (4.4%) samples had two, another four (4.4%) had three alterations, and one patient had four alterations detected (Figure 2).

More than half of the detected mutations 51.8% (29/56) were found in the HGSOC group. In this group, missense and nonsense mutations of *TP53* were found in 10 out of 37 cases (27%). In addition, in 13 (35.1%) cases, *BRCA1* mutations were detected, and in four cases, *BRCA2* mutations (10.8%) were detected. One patient had a missense *PIK3CA* mutation. In uterine lavage samples from other groups (type I OC, EC, benign gynecologic malignancies, and RRS), no *TP53* mutations were detected, including the case with the STIC diagnosis as well.

No alterations were detected in uterine lavage samples from patients with benign gynaecologic conditions. However, in the group of patients with type I OC, one case out of nine (a patient with a serous borderline tumor) had a *KRAS* mutation. Meanwhile, 4/12 (33.3%) endometrial cancer cases had alterations in PI3K pathway genes (*PIK3CA, PTEN,* and *KRAS*). Three patients with mutations in all three PI3K pathway genes had grade 2 endometrioid endometrial carcinoma, while the patient with only *PIK3CA* and *PTEN* mutations was diagnosed with grade 1 endometrial cancer.

In the study cohort, 40 patients underwent genetic consulting and germline *BRCA1/2* testing. Alterations detected in uterine lavage samples were perfectly concordant with the genetic testing results. All 13 RRS and 14 HGSOC patients harbouring germline *BRCA1/2* mutations were detected with the same mutation in uterine lavage samples, while all 13 patients (both HGSOC and other cases) with negative germline *BRCA1/2* testing results also showed no mutations in uterine lavage samples.

### 3.2. Mutation Analysis in Ovarian Tissue Samples

In addition to uterine lavage samples, mutation analysis was also carried out on the 46 tissue samples available (Figure 3). 46 mutations were detected in 67.4% (31/46) of the samples. *TP53* alterations were detected in 79.3% (23/29) of HGSOC patients. One *TP53* mutation was found in one clear-cell OC case.

Multiple mutations were detected in 93.1% (27/29) of HGSOC patients’ tissue samples: *TP53* in 79.3% (23/29), *BRCA1* in 34.4% (10/29), *BRCA2* in 13.7% (4/29), and *PIK3CA* in 6.8% (2/29). In HGSOC patient’s tissue samples without *TP53* mutations, one mutation in *BRCA1*, two mutations in *BRCA2*, and one *PIK3CA* mutation were detected.

In cases other than HGSOC, *BRCA2* alteration was detected in the only available RRS group tissue sample, and four PI3K pathway gene mutations were detected in other ovarian malignant tumor tissue samples. No mutations were detected in the benign gynecologic tumor group.

Overall, the concordance rate between uterine lavage and tissue samples was 65.2% (30/46). However, the positive concordance rate (the amount of fully concordant cases/all mutated tissue samples) was 48.5% (15/31) (Table 2).

Among disconcordant cases, *TP53* mutations were predominant: 13 mutations were detected in 14 tissue samples (*TP53* c.659A > G missense mutation was detected in 2 different tissue samples), which were not detected in uterine lavage (Appendix A).

### 3.3. Uterine Lavage Mutation Correlation with Clinical Features

We further analysed the association between uterine lavage mutation and clinical features. Overall, significantly more mutations were detected in uterine lavage samples from patients with FIGO stages III and IV (*p* = 0.002, Fisher’s exact test) when compared with patients with lower stages of OC or endometrial cancer. In HGSOC, significantly higher proportion of uterine lavage mutations were detected in FIGO stage IV (*p* = 0.002, Fisher’s exact test). The concentration of CA125 for prognostic purposes measured before the treatment was significantly higher in patients with mutations in uterine cavity lavage (*p* = 0.0001, Mann-Whitney test) (Figure 4), while there was no association between mutation status and postsurgical CA125 concentration (*p* = 0.16, Mann-Whitney test).

Besides, in the HGSOC group, patients with *BRCA1* or *BRCA2* alterations were significantly younger than other patients (mean difference 7.0 years, *p* = 0.0183, Welch’s *t* test).

### 3.4. Diagnostic and Predictive Value of Uterine Lavage Mutations

For the analysis of diagnostic parameters of the uterine lavage test, the mutation rate in uterine lavage from HGSOC was compared with that of other study groups (type I OC, endometrial cancer, and benign gynaecologic malignancies), except for the RSS group. In our study cohort, *TP53* uterine lavage mutations were able to detect HGSOC with 27.0% sensitivity and 100% specificity, while the combination of *TP53* and/or *BRCA1/2* uterine lavage-detectable mutations was able to identify HGSOC with 62.2% sensitivity and 100% specificity (Table 3).

Association analysis revealed that the presence of *TP53* mutation in uterine lavage significantly (*p* = 0.0003) increased the risk of HGSOC (Table 4). In uterine lavage, *BRCA1* mutation alone, *TP53* and/or *BRCA1/2* mutations, or mutations in at least one of the four genes in the gene panel were highly (*p* < 0.0001) predictive for HGSOC. The best predictor (risk ratio 3.9) of HGSOC was the presence of *TP53* and/or *BRCA1/2* mutations in uterine lavage.

### 3.5. Prognostic Value of Uterine Lavage Mutations

We then examined whether the uterine lavage mutation could predict the disease progression of the HGSOC group. A subset of 24 HGSOC cases was used in the analysis (patients with stages IIIC and IV.) For this, a Kaplan-Meier curve analysis of PFS stratified by every gene or gene group of interest was performed (Figure 5). Cases with the *TP53* mutation in uterine lavage had a shorter PFS than cases without the mutation (HR = 3.21, 95% CI: 0.73–14.1, 219 *p* = 0.12, univariate Cox regression analysis), though the effect was not statistically significant. Mutations in *BRCA1* or *BRCA2* or the combination of any 4-gene mutations was not predictive for PFS (Figure 5B). In the multivariate Cox regression model adjusted for clinical features (cytoreductive surgery score, FIGO stage, and age), *TP53* mutation in uterine lavage was not significantly associated with PFS.

## 4. Discussion

This proof-of-concept study showed the feasibility of suing a uterine lavage sample as a liquid biopsy for gynecologic cancers. Our pilot study demonstrates that cells shed from Müllerian duct cancer can be collected in uterine lavage, where tumor-specific mutations can be detected through NGS. We focused on *TP53* mutation analysis because HGSOC is characterized by a high frequency of *TP53* mutations, mostly found in all type II OC.

All 90 uterine lavage samples in our study had a sufficient amount of DNA and were successfully analyzed by NGS. Mutations in uterine lavage samples were detected in 62% (23/37) of HGOSC, with predominant alterations of the *TP53* and *BRCA1/2* genes. No such mutations were detected in uterine lavage in type I OC, endometrial carcinomas, or benign gynecological cases, showing a high specificity of selected biomarkers for type II OC. In comparison, in the E. Maritschnegg et al. study [16] 60% of OC patients were identified with *TP53* and other gene mutations in uterine cavity lavage samples.

In our study, *TP53* mutation analysis using the standard NGS technique achieved high specificity, but it lacked the desirable clinical sensitivity. This could be improved by using a more accurate sequencing technique, a different type of liquid biopsy sample type, or an expanded biomarker panel. In the same pilot study by E. Maritschnegg et al. [17], SafeSeqS sequencing and digital droplet polymerase chain reaction (ddPCR) techniques improved mutation detection rates by an additional 20% compared with conventional NGS. A later study from the same group [18,19] applied an ultra-accurate duplex sequencing technique to 10 uterine lavage samples from OC patients and demonstrated similarly high sensitivity (80%) of *TP53* mutation detection but also detected low-frequency *TP53* mutations in nearly all lavages from patients without cancer. These cancer-like *TP53* mutations were highly associated with age. In general, using novel detection methods with high sensitivity for detecting mutations can lead to the discovery of naturally occurring yet very low-frequency (<0.01%) mutations in healthy individuals that may finally result in a false-positive diagnosis. Thus, reasonable balance between sensitivity and specificity of liquid biopsy-based tests should be maintained with a great attitude in cancer diagnostics.

Although uterine lavage analysis in our study “missed” 14 of 24 cases with *TP53* mutations detected in OC tissues, this resulted in a low overall concordance rate of 69.6%. Other OC studies also showed some disconcordance between liquid biopsy and tumor mutations. Jiang et al. study [21] applied circulating single-molecule amplification and resequencing technology (cSMART) to 17 tumors, 11 Pap smears, and 22 plasma samples from OC patients. Although all liquid biopsy samples were positive for OC-related mutations, the concordance rate between liquid biopsy and tumor mutations was 50% for Pap-smear and 71,4% for plasma samples. A recent study [22] analyzing blood-derived ctDNA and tissue *TP53* mutations in patients with various cancers found that *TP53* was fully concordant in 45% (116/258) of cases with the mutations, a similar positive concordance rate was observed in our study (41.7%).

In addition to frequent *TP53* mutations in our study, 46% of HGSOC cases had *BRCA1* or *BRCA2* mutations in uterine lavage, and 82% of them were proven to be germline. Moreover, all patients in the RRS group (familial breast-ovarian cancer syndrome cases) also had germline *BRCA1/2* mutations detected in uterine lavage samples. Typically, *BRCA1/2* mutations are detected in 22% of HGSOC tumors [7]. *BRCA1/2* mutations mainly serve as a prognostic biomarker in HGSOC, while the combination of other biomarkers, such as circulating microRNAs and DNA methylation-based biomarkers, might improve the diagnostic potential of liquid biopsy for early OC detection [23].

One of the limitations of our study was the predominance of FIGO stage III-IV patients in the HGSOC group. In our pilot study, we included patients with ovarian tumors and performed uterine lavage before surgery, when histology was not known. After histology was revealed, I staged OC patients appeared to be type I OC or borderline tumours. To detect HGSOC in FIGO stage I-II is more luck than an accurate diagnostic test. Only one *KRAS* mutation was detected in uterine lavage in a stage I borderline OC case (in 1/7 other OC cases). Studies that included early-stage OC patients [6,18,24] showed poor mutation detection rates in liquid biopsy samples. In both Maritschnegg et al. [6] and Kindle et al. [24] studies, two out of four stage I OC patients had detectable mutations in uterine lavage and Pap smears respectably, and PapSEEK study showed similar low specificity of 34% in both early (I-II) and late (III-IV) OC cases [18].

Currently, the most conventional approach for liquid biopsy use in cancer biomarker research is mutation detection in blood plasma. However, blood has a very low ctDNA level when compared to non-cancerous cfDNA fractions [25]. Inspired by cytological analysis routinely used for the detection of precancerous lesions or early cervical cancer and the possibility to detect ctDNA from endometrial or ovarian cancers in PAP smears, the uterine lavage technique was developed.

In our study, we focused on HGSOC type because type I OC can be detected by conventional ultrasound check up and gynecological examination, eliminating the need for additional early diagnostic tools. Our study proved again that uterine lavage is an efficient technique for gynecologic cancer detection. In addition, our data indicate that uterine lavage and our selected genes are not sensitive enough for early HGSOC detection. Further studies with a larger set of genes or biomarker combinations (miRNA, DNR methylation, and others) in larger independent cohorts are needed for reliable diagnostic test development. However, our study provided proof of principle for the suitability of uterine lavage samples for the development of such diagnostic tests.

## 5. Conclusions

This study proved that ctDNA from ovarian neoplasms can be collected via lavage of the uterine cavity for diagnostic needs in an outpatient setting. Our study revealed precise detection of *TP53, BRCA1, BRCA2,* and other gene mutations in uterine lavage from HGSOC by means of NGS. However, for improved sensitivity of such test, additional disease-specific biomarkers have to be discovered.

## Figures and Tables

**Figure 1 cancers-15-00868-f001:**
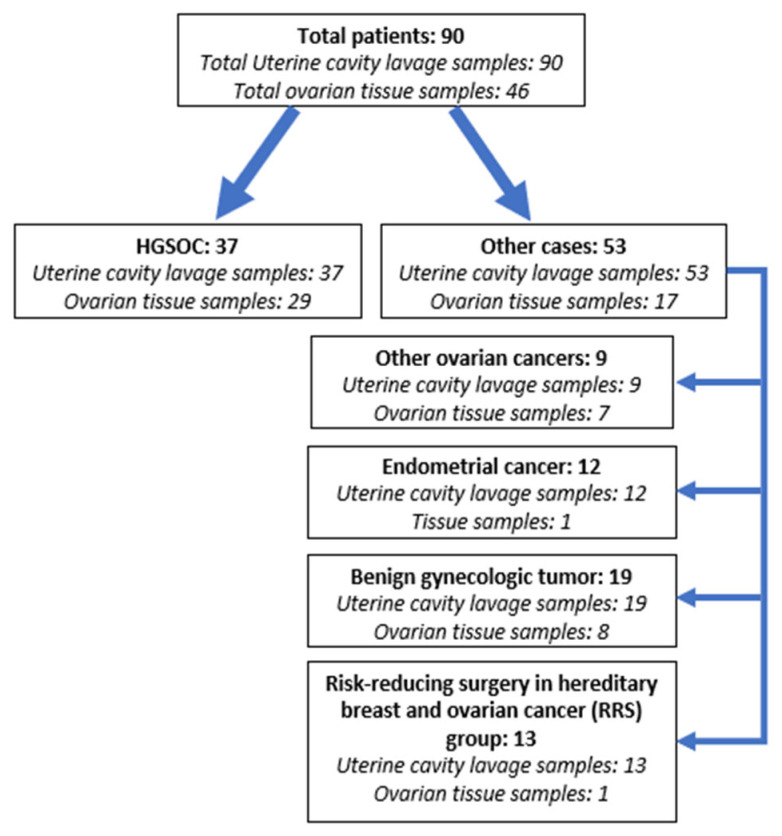
Schematic representation of the study cohort.

**Figure 2 cancers-15-00868-f002:**
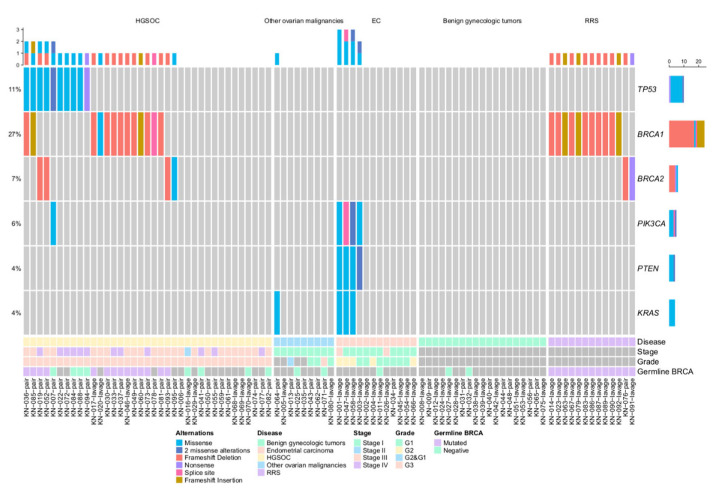
OncoPrint of uterine cavity lavage samples. EC–endometrial cancer.

**Figure 3 cancers-15-00868-f003:**
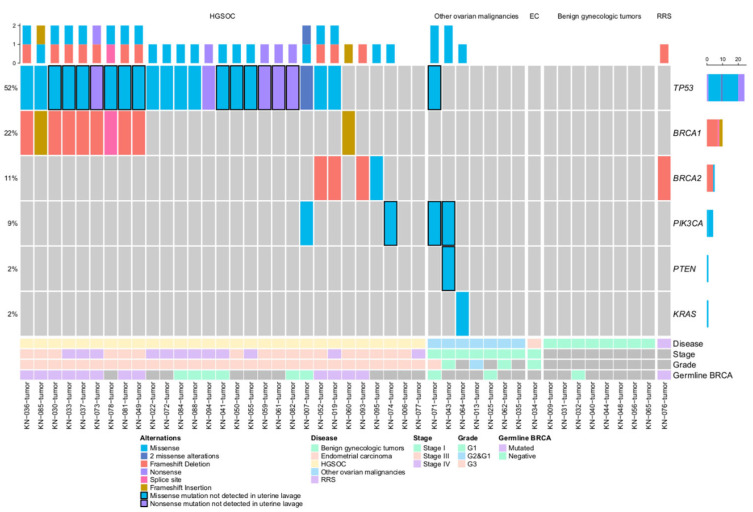
OncoPrint of ovarian tissue samples, EC—endometrial cancer. The black border indicates mutations not found in uterine lavage samples.

**Figure 4 cancers-15-00868-f004:**
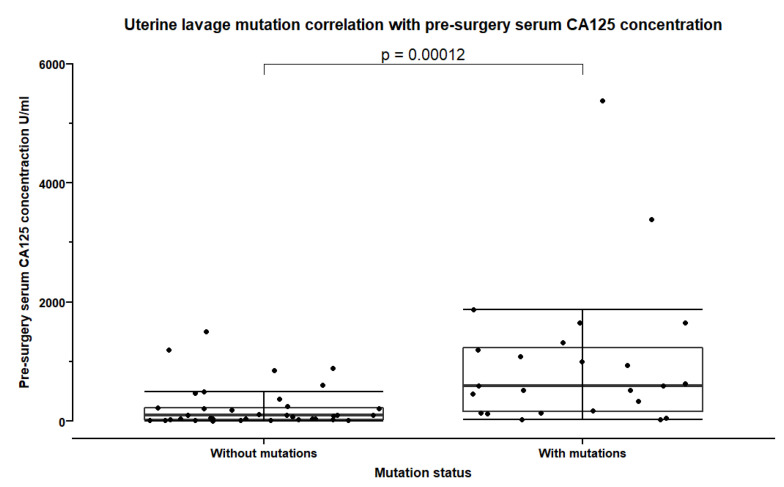
Pre-surgery serum CA125 concentration in patients with and without detected mutations in uterine cavity lavage samples. Error bars show the mean and standard deviation.

**Figure 5 cancers-15-00868-f005:**
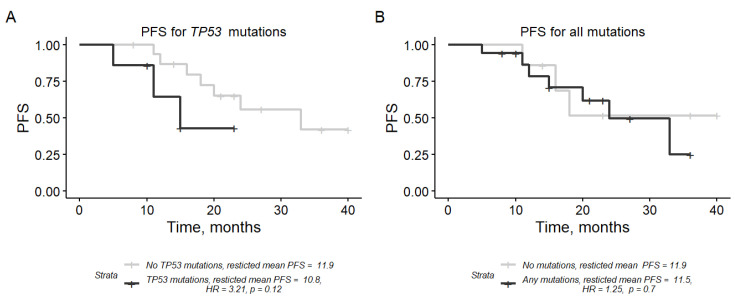
Kaplan-Mayer curves for progress-free survival (PFS) at 12 months stratified by uterine lavage mutations in (**A**) *TP53* and (**B**) all genes. Restricted mean PFS is denoted in months. HR—hazard ratio, CI—confidence interval.

**Table 1 cancers-15-00868-t001:** Clinical-pathological characteristics of the study cohort.

Disease Group	HGSOC (%)	Other Ovarian Cancers (%)	Endometrial Cancer (%)	Benign Gynecologic Tumor (%)	RRS Group (%)	Overall (%)
*n* =	37	9	12	19	13	90
Average Age, years (min-max)	58.2 (42–82)	62.6 (49–75)	62.8 (56–74)	55.9 (41–83)	46.3 (35–65)	57.0 (35–83)
Average CA125 pre-operative concentration U/mL (N/A)	848.7 (1 N/A)	152.1 (3 N/A)	25.3 (10 N/A)	51.5 (2 N/A)	20.0 (12 N/A)	538.5 (29 N/A)
FIGO Stage						
IA		8 (88.9)	6 (50.0)			14 (15.6)
IB		1 (11.1)	4 (33.3)			5 (5.6)
IIB	1 (2.7)					1 (1.1)
IIIA	1 (2.7)					1 (1.1)
IIIB	4 (10.8)					4 (4.4)
IIIC	19 (51.4)		2 (16.7)			21 (23.3)
IVB	12 (32.4)					12 (13.3)
N/A ^1^				19 N/A	13 N/A	32 (35.6)
Tumour differentiation grade						
G1		3 (33.3)	7 (58.3)			10 (11.1)
G2		1 (11.1)	5 (41.7)			6 (6.7)
G3	37 (100.0)	1 (11.1)				38 (42.2)
BD ^2^		3 (33.3)				3 (3.3)
N/A ^1^		1 (11.1)		19 (100.0)	13 (100.0)	33 (36.7)
Progressed disease						
Yes	18 (48.7)		1 (8.3)			12 (13.3)
No	19 (51.4)	9 (100.0)	11 (91.7)			46 (51.1)
N/A				19 N/A	13 N/A	32 (35.6)
Deceased						
Yes	5 (13.5)		1 (8.3)			6 (6.7)
No	32 (86.5)	9 (100.0)	11 (91.7)			52 (57.8)
N/A ^1^				19 N/A	13 N/A	32 (35.5)
Mutation status (uterine lavage samples)						
*TP53*	10 (27.0)					10 (11.1)
*BRCA1*	13 (35.1)				11 (84.6)	24 (26.6)
*BRCA2*	4 (10.8)				2 (15.4)	6 (6.6)
*PI3KCA*	1 (2.7)		4 (33.3)			5 (5.5)
*PTEN*			4 (33.3)			4 (4.4)
*KRAS*		1 (11.1)	3 (25.0)			4 (4.4)
Mutation status (ovarian tissue samples)						
*TP53*	23 (79.3)	1 (14.2)				24 (52.2)
*BRCA1*	10 (34.4)					10 (21.7)
*BRCA2*	4 (13.8)					4 (8.7)
*PI3KCA*	2 (6.8)	2 (28.6)				4 (8.7)
*PTEN*		1 (14.2)				1 (2.2)
*KRAS*		1 (14.2)				1 (2.2)

^1^ N/A—no data, ^2^ BD—borderline.

**Table 2 cancers-15-00868-t002:** Overall and positive concordance rates and Kappa values describing agreement categories (1 = perfect agreement, 0 = total disagreement). Overall concordance is calculated as (++)+(− −)/total cases, while positive concordance rate is calculated as (++)/((++)+(+ −)).

		Tissue		Overall Concordance Rate %	Positive Concordance Rate %	Kappa (SE)
	ctDNA	+	−			
*TP53*	+	10	0	69.565	41.667	0.406 (0.107)
	−	14	22			
*BRCA1/2*	+	15	0	100.000	100.000	1(0)
	−	0	31			
*PI3K* pathway	+	2	0	91.304	33.304	0.465 (0.216)
	−	4	40			
Any mutation	+	15	0	65.217	48.487	0.379 (0.098)
	−	16	15			

**Table 3 cancers-15-00868-t003:** Performance characteristics of uterine lavage mutations for type II ovarian cancer diagnosis comparing type II vs. controls, without the RRS group. PPV—positive predictive value, NPV—negative predictive value.

Performance of HGSOC vs. other Cases Except for the RSS Group	Sensitivity%	Specificity%	Accuracy%	PPV%	NPV%
*TP53*	27.03	100.0	64.94	100.0	59.70
*BRCA1/2 + TP53*	62.16	100.0	81.82	100.0	74.07
*BRCA1*	35.14	100.0	68.83	100.0	62.50
*BRCA2*	10.81	100.0	57.14	100.0	54.79
PI3K pathway mutations	2.70	87.50	46.75	16.67	49.30
Any gene mutation	62.16	87.50	75.32	82.14	71.43

**Table 4 cancers-15-00868-t004:** Predictive value of uterine lavage mutations for type HGSOC diagnosis (comparing type II vs. controls, without the RRS group). OR—odds ratio.

Predictive Risk of HGSOC vs. Other Cases Except the RSS Group	Risk Ratio	Risk Ratio 95% CI	OR (Fishers Test)	OR 95% CI	Fishers Test, *p*-Value
*TP53*	2.481	1.854–3.321	INF	2.956-INF	0.0003
*BRCA1/2 + TP53*	3.857	2.457–6.054	INF	13.444-INF	<0.0001
*BRCA1*	2.667	1.944–3.659	INF	4.456-INF	<0.0001
*BRCA2*	2.212	1.718–2.847	INF	0.740-INF	0.0488
PI3K pathway mutations	0.329	0.0542–1.996	0.198	0.004–1.898	0.2022
Any gene mutation	2.875	1.788–4.624	11.072	3.287–45.033	<0.0001

## Data Availability

Not applicable.

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
