# Peer review of "Uterine Cavity Lavage Mutation Analysis in Lithuanian Ovarian Cancer Patients"

_cancers, 2023, doi:10.3390/cancers15030868_

Round 1

Reviewer 1 Report

The authors investigated the useful of uterine lavage for detection of type ovarian cancer (OC). Uterine lavage is a lowly invasive, simple test that is very interesting as a diagnostic method. My comments are as follow.

(1) The aim of this study is to examine whether the detection of TP53 mutation in uterine cavity lavage can be used as a diagnostic method for HGSOC. However, it is less significant to identify the histological type by uterine lavage in patients who have already developed ovarian cancer. 

(2) The low concordance rate between tissue and uterine lavage makes it impossible to determine a treatment strategy, such as whether to administer PARP inhibitors based on genetic changes in the uterine lavage.

(3) Uterine lavage can provide information on HRD-related genes other than BRCA. The degree of abnormality should be presented.

(4) The authors described effective screening approaches for OC in the Introduction Section. However, the data do not indicate that uterine lavage is useful in the early detection of OC. It would be very interesting to investigate if a study can predict the development of ovarian cancer by uterine lavage in a population of healthy subjects. Also, TP53 mutation was not detected in the RRSO cases and BRCA mutation was detected in all cases. The fact that TP53 is the target gene for early detection is questionable.

Author Response

Dear Editors-In-Chief,

We want to thank the Reviewers for their constructive comments and efforts towards improving our manuscript. We are grateful for insightful comments on our paper and we have been able to incorporate changes to reflect most of the suggestions provided. Please find below a detailed point-by-point response to the comments of the Reviewers. The main corrections made in the manuscript text have been highlighted.

(1) The aim of this study is to examine whether the detection of TP53 mutation in uterine cavity lavage can be used as a diagnostic method for HGSOC. However, it is less significant to identify the histological type by uterine lavage in patients who have already developed ovarian cancer.

Our pilot study aimed to detect whether uterine lavage is suitable and specific for HGSOC detection throw ctDNA TP53 mutation analysis. Our study goal wasn’t to identify the histological type by uterine lavage. In our study we involved patients who underwent surgery due to diagnosed ovarian tumor, and prior to uterine lavage sampling before surgery, histological type of the tumor wasn’t know. HGSOC is aggressive OC type and usually diagnosed at III-IV FIGO stage, so to catch I-II stage is a challenge. In our study we focused in this type of OC, because type I OC is mostly diagnosed at first stages and not so aggressive. Conventional ultrasound check up and gynecological examination can determine type I OC at early stages, there is no need for early diagnostic tool.

(2) The low concordance rate between tissue and uterine lavage makes it impossible to determine a treatment strategy, such as whether to administer PARP inhibitors based on genetic changes in the uterine lavage.

BRCA1/2 mutations mainly serve as a prognostic biomarker in HGSOC as the cases with such mutations present in superior median overall survival and sensitivity to poly (ADP-ribose) polymerase inhibitor (PARP) treatment. All cases with known germline BRCA1/2 mutations were identified with the mutation in uterine lavage as well. The main technical problem was related to TP53 mutation detection in uterine lavage. As the main aim of our study was the development of diagnostic, not prognostic test, the sentence was taken out.

(3) Uterine lavage can provide information on HRD-related genes other than BRCA. The degree of abnormality should be presented.

In this pilot study we attempted to develop the test for ovarian cancer specific changes detection in uterine lavage DNA. In this study 6 ovarian cancer-related genes (TP53, BRCA1, BRCA2, PIK3CA, PTEN and KRAS) were included and mutation analysis performed by NGS. All detected abnormalities are presented in Fig.2 and 3. We agree with the reviewer that a larger set of HRD-related genes can be included in further studies.

(4) The authors described effective screening approaches for OC in the Introduction Section. However, the data do not indicate that uterine lavage is useful in the early detection of OC. It would be very interesting to investigate if a study can predict the development of ovarian cancer by uterine lavage in a population of healthy subjects. Also, TP53 mutation was not detected in the RRSO cases and BRCA mutation was detected in all cases. The fact that TP53 is the target gene for early detection is questionable.

We agree with the reviewer’s comment. Our data indicate that uterine lavage and selected genes are not sensitive enough for early HGSOC detection. Further studies with a larger set of genes or biomarker combination (miRNA, DNR methylation and others) in larger independent cohorts are needed for reliable diagnostic test development. However, our study provided proof off principle for suitability of uterine lavage samples for such diagnostic test development. According to HGSOC pathogenesis TP53 somatic mutation can be found in precancerous lesion (STIC) in 98-99% of the cases. RRSO group had only one patient with STIC, other didn’t have precancerous lesion.

Reviewer 2 Report

Comments to the Author

The manuscript entitled, “Uterine Cavity Lavage Mutation Analysis in Lithuanian Ovarian Cancer Patients” is worth acceptance for publication in Cancers, however I think it need a major revision to be publish.

One big issue for this manuscript is samples of lavage from endometrial cancer patient were included in the investigation. It complicates and weaken the point of early detection of ovarian cancer. So, I think these cases should be excluded from the analysis.

The following points are what I am concerned about to be revised.

  1. The main point of this manuscript is the early detection of ovarian cancer by analyzing uterine cavity lavage. I think the samples from endometrial cancer patients will not be needed in this investigation and they should be excluded. Uterine cavity lavage from endometrial cancer cases will contain cancer cells and DNA directly. It will be apart from the purpose of the study.
  2. Type II ovarian cancer should be written as high-grade serous ovarian cancer (HGSOC) instead.
  3. Is it correct that ctDNA in the uterine cavity lavage was used for the analysis? Then what did the author use stored uterine lavage cell pellet for the experiment? In the conclusion section, the author said ‘this study proved that tumor cells from ovarian neoplasms…’ that indicate the cells from lavage were used for analysis. While ctDNA was represented in table 2.

In the case of BRCA mutation who underwent RRS, all the cells may contain BRCA mutations. For this point, how can we distinguish ctDNA from endometrial cells to fallopian tube cells? I thought that BRCA mutations detected from RRS cases are just from the ctDNA of endometrial cells. It would be the same as the detection of BRCA mutation in ovarian cancer patients with BRCA germline mutation. 

Author Response

Dear Editors-In-Chief,

We want to thank the Reviewers for their constructive comments and efforts towards improving our manuscript. We are grateful for insightful comments on our paper and we have been able to incorporate changes to reflect most of the suggestions provided. Please find below a detailed point-by-point response to the comments of the Reviewers. The main corrections made in the manuscript text have been highlighted.

  1. The main point of this manuscript is the early detection of ovarian cancer by analyzing uterine cavity lavage. I think the samples from endometrial cancer patients will not be needed in this investigation and they should be excluded. Uterine cavity lavage from endometrial cancer cases will contain cancer cells and DNA directly. It will be apart from the purpose of the study.

Thank you for your comments, we appreciate the time and effort that you have dedicated to providing your valuable feedback on our manuscript. The cases with endometrial cancer were included as a control, to detect HGSOC-specific profile of genetic mutations. Also, to check TP53 mutation detection rate in endometrial group, as accorgind to literature TP53 in EC group account about 10%. As it is shown in fig.2, the mutation profile in uterine lavage from of endometrial cancer cases and HGSOC cases is absolutely different. In further studies of HGSOC-specific mutations no more such control will be needed.

  1. Type II ovarian cancer should be written as high-grade serous ovarian cancer (HGSOC) instead.

According to reviewer’s suggestion the type II OC was renamed to HGSOC.

  1. Is it correct that ctDNA in the uterine cavity lavage was used for the analysis? Then what did the author use stored uterine lavage cell pellet for the experiment? In the conclusion section, the author said ‘this study proved that tumor cells from ovarian neoplasms…’ that indicate the cells from lavage were used for analysis. While ctDNA was represented in table 

Cell free DNA derived from cancerous and other cells were used in our study. While the mutation detection assisted in identification of cancer cell-derived ctDNA. For clarity, according to reviewer’s comment we included ctDNA instead of cancer cell in the conclusion section of the manuscript. The resulting uterine lavage cell pellet was resuspended in 2 mL PBS and stored at -80°C until use. DNA was dissolved in nuclease free water, and stored at -20°C until further steps.

  1. In the case of BRCA mutation who underwent RRS, all the cells may contain BRCA mutations. For this point, how can we distinguish ctDNA from endometrial cells to fallopian tube cells? I thought that BRCA mutations detected from RRS cases are just from the ctDNA of endometrial cells. It would be the same as the detection of BRCA mutation in ovarian cancer patients with BRCA germline mutation.We agree with the Reviewer’s comment, that in the cases with hereditary BRCA mutation all cells contain BRCA mutation. RRS cases in our study was included as a controls, while the main object of the study was HGSOC cases. RRS group was important as we wanted to check TP53 mutation status in precancerous lesion, but only one patient in RRS group had STIC and no mutation ere found. ctDNA compose only a small fraction of all lavage derived cell-free DNA and is distinguished by the presence of tumor-specific mutations. Our study revealed TP53, and BRCA mutations as specific for HGSOC, but further studies with a large set of genes in independent cohorts are needed in order to increase diagnostic potential of uterine lavage test.

Round 2

Reviewer 1 Report

The authors' responses are satisfactory, but they are not reflected in the revised manuscript. Could you added and stated them in the Discussion Section?

Author Response

Thank you for your comments, we reflected and added them now in discussion section. We have highlighted the changes within the manuscript in yellow.

Thank you for your effort. 

Reviewer 2 Report

I think the author's reply is well and the revised manuscript is worth acceptance.

Author Response

Thank you, we appreciate the time and effort that you have dedicated. We have highlighted the changes within the manuscript in yellow.
